



# Role of OH variability in the stalling of the global atmospheric CH$_4$ growth rate from 1999 to 2006

**J. McNorton[1,2], M. P. Chipperfield[1,2], M. Gloor[3], C. Wilson[1,2], W. Feng[1,4],**

**G. D. Hayman[5], M. Rigby[6], P. B. Krummel[7], S. O'Doherty[6], R. G. Prinn[8], R. F. Weiss[9],**

**D. Young[6], E. Dlugokencky[10], and S. A. Montzka[10]**

1. School of Earth and Environment, University of Leeds, Leeds, LS2 9JT, UK.

2. National Centre for Earth Observation, University of Leeds, LS2 9JT, UK.

3. School of Geography, University of Leeds, Leeds, LS2 9JT, UK.

4. National Centre for Atmospheric Science, University of Leeds, LS2 9JT, UK.

5. Centre for Ecology and Hydrology, Wallingford, UK.

6. School of Chemistry, University of Bristol, Bristol, BS8 1TS, UK.

7. CSIRO Oceans and Atmosphere Flagship, Aspendale, Victoria, Australia.

8. Center for Global Change Science, Massachusetts Instititute of Technology, Cambridge, MA 02139, USA.

9. Scripps Institution of Oceanography, University of California, San Diego,CA 92093, USA.

10. National Oceanic and Atmospheric Administration, Boulder, USA.

## Abstract

The growth in atmospheric methane (CH$_4$) concentrations over the past two decades has shown large variability on a timescale of many years. Prior to 1999 the globally averaged CH$_4$ concentration was increasing at a rate of 6.0 ppb/yr, but during a stagnation period from 1999 to 2006 this growth rate slowed to 0.6 ppb/yr. Since 2007 the growth rate has again increased to 4.9 ppb/yr. These changes in growth rate are usually ascribed to variations in CH$_4$ emissions. We have used a 3-D global chemical transport model, driven by meteorological reanalyses and variations in global mean hydroxyl (OH) concentrations derived from CH$_3$CCl$_3$ observations from two independent networks, to investigate these CH$_4$ growth variations. The model shows that between 1999 and 2006, changes in the CH$_4$ atmospheric loss contributed significantly to the suppression in global CH$_4$ concentrations relative to the pre-1999 trend. The largest factor in this is relatively small variations in global mean OH on a timescale of a few years, with minor contributions of atmospheric transport of CH$_4$ to its sink region and atmospheric temperature. Although changes in emissions may be important during the stagnation period, these results imply a smaller variation is required to explain the observed CH$_4$ trends. The contribution of OH variations to the renewed CH$_4$ growth after 2007 cannot be determined with data currently available.



## 1. Introduction


The global mean atmospheric methane ($CH_4$) concentration has increased by a factor of 2.5
since the pre-industrial era, from approximately 722 ppb in 1750 to $1803.2 \pm 0.7$ ppb in 2011
(Etheridge et al., 1998; Dlugokencky et al., 2005). Over this time period methane has accounted
for approximately 20% of the total direct anthropogenic perturbation of radiative forcing by
long-lived greenhouse gases ($0.48\pm0.05$ W/m$^2$), the second largest contribution after $CO_2$
(Cicerone et al., 1988; Myhre et al., 2013). This long-term methane increase has been attributed
to a rise in anthropogenic emissions from fossil fuel exploitation, agriculture, waste
management and biomass burning (Dlugokencky et al., 2011). Predictions of future $CH_4$ levels
require a complete understanding of processes governing emissions and atmospheric removal.
Since the mid-1980s measurements of $CH_4$ in discrete atmospheric air samples collected at
surface sites have been used to observe changes in the interannual growth rate of $CH_4$ (Rigby
et al., 2008; Dlugokencky et al., 2011, Kirschke et al., 2013). Nisbet et al. (2014) showed that
between 1984 and 1992 atmospheric $CH_4$ increased at ~12 ppb/yr, after which the growth rate
slowed to ~3 ppb/yr. In 1999 a period of near-zero growth began which continued until 2007.
In 2007 this stagnation period ended and since then average growth has increased again to ~6
ppb/yr (Rigby et al., 2008; Dlugokencky et al., 2011).
The reasons for the pause in $CH_4$ growth are not well understood. Bousquet et al. (2006)
performed an atmospheric transport inversion study to infer an increase in anthropogenic
emissions since 1999. Similarly, the EDGAR v3.2, bottom-up anthropogenic emission
inventory, an updated inventory to that used as an a priori by Bousquet et al. (2006), shows a
year-on-year increase in anthropogenic $CH_4$ emissions between 1999 and 2006 (Olivier et al.,
2005). This would suggest that a decrease in anthropogenic emissions is not the likely cause of
the pause in growth during this period. A second explanation is a reduction in wetland
emissions between 1999 and 2006, which is in part compensated by an increase in
anthropogenic emissions (Bousquet et al., 2006). However, more recently, Pison et al. (2013)
used two atmospheric inversions and found much more uncertainty in the role wetlands played
in the pause in growth over this period.
Dlugokencky et al. (2003) argued that the behaviour of global mean $CH_4$ up to around 2002
was characteristic of the system approaching steady state, accelerated by decreasing emissions
at high northern latitudes in the early 1990s and fairly constant emissions elsewhere. However,
since then there have been notable perturbations to the balance of sources and sinks (Rigby et
al., 2008). This has been, at least partly, attributed to increases in wetland (Bousquet et al.,
2011) and anthropogenic emissions (Bousquet et al., 2006). Recent changes in emissions are
not well constrained and the reasons for the renewed growth are also not fully understood.
Atmospheric chemistry has also been hypothesised to play a role in past variations in $CH_4$
growth rates. The major (90%) sink of atmospheric $CH_4$ is via reaction with the hydroxyl
radical, OH. Variations in the concentration of OH ([OH]), or changes to the reaction rate
through changes in temperature, therefore have the potential to affect $CH_4$ growth. Previous
studies have suggested that an increase in atmospheric OH concentration may have been at
least partly responsible for a decrease in the $CH_4$ growth rate (Lelieveld et al., 2004; Fiore et





al., 2006). This rise in OH has been attributed to an increase in lightning $NO_x$ (Fiore et al., 2006). The abundance of other species such as $H_2O$, $O_3$, CO and $CH_4$ also determine the concentration of OH (Leliveld et al., 2004). Prinn et al. (2005) suggested that major global wildfires and El Nino Southern Oscillation (ENSO) events could influence [OH] variability. A reduced number of small- to moderate-magnitude volcanic eruptions during the $CH_4$ stagnation period (Carn et al., 2015; Mills et al., 2015) may have increased [OH], due to increased downward UV radiation. Recently, Patra et al. (2014) analysed global $CH_3CCl_3$ observations for 2004-2011 to derive the interhemispheric ratio of OH. In contrast to many model results which suggest higher mean [OH] in the north, they derived similar values for both hemispheres.

Warwick et al. (2002) investigated the impact of meteorology on atmospheric $CH_4$ growth rates from 1980 to 1998, i.e. well before the observed recent pause. They concluded that atmospheric conditions could be an important driver in the interannual variability (IAV) of atmospheric $CH_4$. In similar studies a combination of atmospheric dynamics and changes in emissions were shown to explain some of the earlier past trends in atmospheric $CH_4$ (Fiore et al., 2006; Patra et al., 2009). This paper builds on these studies to investigate the chemical and non-chemical atmospheric contribution to the recent variations in $CH_4$ growth. By 'non-chemical' we mean transport-related influences, although the loss of $CH_4$ is ultimately due to chemistry as well. We use a global chemical transport model to simulate the period from 1993 to 2011 and to quantify the impact of variations in [OH] and meteorology on atmospheric $CH_4$ growth.

## 2. Data and Models

### 2.1 NOAA and AGAGE $CH_4$ Data and Derived OH

We have used surface $CH_4$ observations from 19 National Oceanographic and Atmospheric Administration/Earth System Research Laboratory (NOAA/ESRL) cooperative global air sampling sites (Dlugokencky et al., 2014) over 1993-2009 (see Table 1). To calculate the global average concentration, measurements were interpolated across 180 latitude bins, which were then weighted by surface area. We have also used the same method to derive global mean $CH_4$ based on 5 sites from the Advanced Global Atmospheric Gases Experiment (AGAGE) network (Prinn et al., 2000; Cunnold et al., 2002; Prinn et al., 2015).

Montzka et al. (2011) used measurements of methyl chloroform ($CH_3CCl_3$) from an independent set of flasks sampled at a subset of NOAA air sampling sites to derive global [OH] anomalies from 1997 to 2007. They argued that uncertainties in emissions are likely to limit the accuracy of the inferred inter-annual variability in global [OH], particularly before 1997. At this time the emissions were large but decreasing rapidly due to the phaseout of $CH_3CCl_3$ production and consumption, and the large atmospheric gradients were also more difficult to capture accurately with only few measurement sites. Instrument issues caused an interruption to their $CH_3CCl_3$ time series in 2008/9. We have averaged these (based on the red curve in Figure 3 of Montzka et al.) into yearly anomalies to produce relative interannual variations in the mean [OH]. Similarly, Rigby et al. (2013) used $CH_3CCl_3$ measurements from the 5-station AGAGE network in a 12-box model to produce yearly global [OH] anomalies from 1995 (the date from which data from all 5 stations is available) to 2010. These two timeseries, which convert anomalies in the $CH_3CCl_3$ decay rate into anomalies in [OH] using constant



temperature, correspond to the best estimate of [OH] variability from the two measurement
networks by the groups who operate them. We then applied these two series of yearly anomalies
uniformly to the global latitude-height [OH] field used in the recent TransCom CH$_4$ model
intercomparison (see Patra et al., 2011), which itself was derived from a combination of semi-
empirically calculated tropospheric OH distributions (Spivakovsky et al. 2000; Huijnen et al.,
2010) and 2-D model simulated stratospheric loss rates (Velders, 1995). For consistency
between the model experiments, both sets of yearly anomalies were scaled so that the mean
[OH] between 1997 and 2007 (the overlap period where NOAA and AGAGE anomalies are
both available) equalled the TransCom [OH] value. In the rest of this paper we refer to these
two OH datasets as 'NOAA-derived' and 'AGAGE-derived'.
These two calculations of yearly [OH] anomalies use slightly different assumptions for
CH$_3$CCl$_3$ emissions after 2002. Before that date they use values from Prinn et al. (2005). The
NOAA data then assumed a 20% decay in emission for each subsequent year (Montzka et al.,
2011), while AGAGE used United Nations Environment Programme (UNEP) consumption
values (UNEP, 2015). Holmes et al. (2013) suggested that inconsistencies in CH$_3$CCl$_3$
observations between the AGAGE and NOAA networks also limit understanding of OH
anomalies for specific years due to an unexplained phasing difference of up to around 3 months.
As we are interested in the impact of [OH] changes over longer time periods (e.g. 2000 – 2006),
this phase difference will be less important. We have investigated the impact of the different
CH$_3$CCl$_3$ observations and assumed emissions on the derived [OH] anomalies (see Section
136    3.1).

**2.2 TOMCAT 3-D Chemical Transport Model**
We have used the TOMCAT global atmospheric 3-D off-line CTM (Chipperfield, 2006) to
model atmospheric CH$_4$ and CH$_3$CCl$_3$ concentrations. The TOMCAT simulations were forced
by winds and temperatures from the 6-hourly European Centre for Medium-Range Weather
Forecasts (ECMWF) ERA-Interim reanalyses (Dee et al., 2011). They covered the period 1993
to 2011 with a horizontal resolution of 2.8° × 2.8° and 60 levels from the surface to ~60 km.
The TOMCAT simulations use annually repeating CH$_4$ emissions, which have been scaled to
previous estimates of 553 Tg/yr (Ciais et al., 2013), taken from various studies (Fiore et al.,
2006; Curry et al., 2007; Bergamaschi et al., 2009; Pison et al., 2009; Spahni et al., 2011; Ito
et al., 2012). Annually-repeating anthropogenic emissions (except biomass burning) were
calculated from averaging the EDGAR v3.2 (2009) inventory from 1993 to 2009 (Olivier and
Berowski, 2001). Biomass burning emissions were calculated using the GFED v3.1 inventory
and averaged from 1997 to 2009 (van der Werf et al., 2010). The Joint UK Land Environment
Simulator (JULES) (Best et al., 2011; Clark et al., 2011; Hayman et al., 2014) was used to
calculate a wetland emission inventory between 1993 and 2009, which was then used to
produce a scaled mean annual cycle. Annually-repeating rice (Yan et al., 2009), hydrate, mud
volcano, termite, wild animal and ocean (Matthews et al., 1987) emissions were taken from the
TransCom CH$_4$ study (Patra et al., 2011). The methane loss fields comprised an annually
repeating soil sink (Patra et al., 2011), an annually repeating stratospheric loss field (Velders,





1995) and a specified [OH] field. The model was spun up for 15 years prior to initialising the
simulations, using emission data from 1977 where available and annual averages otherwise.
Fifteen TOMCAT simulations were performed each with a $CH_4$ tracer and a $CH_3CCl_3$ tracer.
The runs had differing treatments of meteorology (winds and temperature) and [OH] (see Table
2). Simulations with repeating [OH] fields (RE_xxxx) used the TransCom dataset. The other
runs with varying [OH] used the NOAA-derived or AGAGE-derived [OH] fields based on the
original published work or our estimates (see Section 3.1). For these runs, the mean [OH] field
is used where the respective NOAA or AGAGE-derived [OH] is unavailable or uncertain
(before 1997 / after 2007 for NOAA and before 1997 / after 2009 for AGAGE). The five
simulations with fixed wind and temperature fields (with labels ending in FTFW) used the
ERA-Interim analyses from 1996 repeated for all years. The five simulations with varying
winds and fixed temperature (with labels ending in FTVW) used zonal mean temperature fields
averaged from 1993-2009. The OH anomalies are derived from the anomaly in the $CH_3CCl_3$
loss rate, which combines variations in atmospheric OH concentration with variations in
temperature which affect the rate constant of the $CH_3CCl_3$ + OH reaction. To quantify the
importance of this temperature effect we also performed 5 model runs which allow both winds
and temperature to vary interannually according to ERA-Interim data (labels ending VTVW).
**3. Results**
**3.1 Correlation of $CH_4$ variations with OH and temperature**
We first investigate the extent to which variations in the observed $CH_4$ growth rate correlate
with variations in derived [OH]. Figure 1a shows the published NOAA-derived and AGAGE-
derived global [OH] anomalies along with the annual $CH_4$ growth rate estimated from the
NOAA and AGAGE measurements. The two [OH] series show the similar behaviour of
negative anomalies around 1997 and 2006/7, and an extended period of more positive
anomalies in between. For the time periods covered by the NOAA (1997-2007) and AGAGE
(1997-2009) $CH_3CCl_3$ observations, the two derived [OH] timseries show negative correlations
with the $CH_4$ growth from NOAA (regression coefficient, R = -0.32) and AGAGE (R = -0.64).
Only the AGAGE [OH] correlation, from the longer timeseries, is statistically significant at the
90% level.
We can use a simple 'global box model' (see Supplement S1) to estimate the [OH] variations
required to fit the observed $CH_4$ growth rate variations assuming constant $CH_4$ emissions and
temperature (black line in Figure 1b). This provides a crude guide to the magnitude of OH
variations which could be important for changes in the $CH_4$ budget. Our results are consistent
with those of Montzka et al. (2011) who performed a similar analysis on the NOAA $CH_4$ data.
The required [OH] rarely exceeds their $CH_3CCl_3$-derived interannual variability (IAV) range
of [OH] (±2.3%, shown as shading in the figure). Also shown in Figure 1b are the published
estimates of the global mean OH anomalies from Figure 1a, converted to concentration units
(see Section 2.1). The relative interannual variations in [OH] required to fit the $CH_4$
observations match the $CH_3CCl_3$-derived [OH] variations in many years, for example from
1998-2002 (see Montzka et al., 2011). Some of the derived variations in [OH] exceed that



required to match the CH$_4$ growth rate, with larger negative anomalies in the early and later
years and some slightly larger positive values in middle of the period.
Figures 1c and 1d show our estimates of [OH] using NOAA and AGAGE observations and
two assumptions of post-2000 CH$_3$CCl$_3$ emissions (see Section 2.1) in a global box model. The
figures also compare our OH estimates with the NOAA-derived and AGAGE-derived [OH]
anomalies based on the work of the observation groups (Figure 1a). Our results demonstrate
the small impact of using different observations and post-2000 emission assumptions (compare
filled and open red circles for the two panels). For these box model results there is also only a
very small effect of using annually varying temperature (compare red and blue lines). In later
years the choice of observations has a bigger impact than the choice of emissions on the derived
[OH]. For AGAGE-derived values (Figure 1d) our estimates agree well with the published
values of Rigby et al., (2013), despite the fact we use a global box model while they used a
more sophisticated 12-box model. In constrast, there are larger differences between our values
and the NOAA-derived OH variability published by Montzka et al. (2011) (Figure 1c), despite
both studies using box models. In particular, around 2002-2003 we overestimate the positive
anomaly in [OH]. We also estimate a much more negative OH anomaly in 1997 than Montzka
et al., though we slightly underrestimate the published AGAGE-derived anomaly in that year
(Figure 1d). Tests show that differences between our results and the NOAA box model are due
to the treatment of emissions. This suggests a larger uncertainty in the inferred low 1997 [OH]
value, when emissions of CH$_3$CCl$_3$ were decreasing rapidly, although reasons why atmospheric
[OH] might have been anomously low were discussed by Prinn et al. (2005). In the subsequent
analysis we use the OH variability from the published NOAA and AGAGE studies as input to
the 3-D model.

### 3.2 TOMCAT Simulations

Overall, Figure 1 shows the potential importance of small, observationally derived variations
in OH concentrations to impact methane growth. We now investigate this quantitatively in the
framework of a 3-D CTM.

### 3.2.1 Methyl Chloroform

The TOMCAT simulations include a CH$_3$CCl$_3$ tracer. This allows us to verify that our approach
of using a global OH field, scaled by derived anomalies, allows the model to reproduce the
observed magnitude and variability of CH$_3$CCl$_3$ decay accurately. Figure 2a shows that the
model, with the imposed [OH] field, does indeed simulate the global decay of CH$_3$CCl$_3$ very
well. This justifies our use of the 'offline' [OH] field, as models with interactive tropospheric
chemistry can produce a large range in absolute global mean [OH] and therefore in lifetimes
of gases such as CH$_3$CCl$_3$. For example, Voulgarakis et al., (2013) analysed the global mean
[OH] from various 3D models and found a range of $0.55 \times 10^6$ to $1.34 \times 10^6$ molecules cm$^{-3}$.
Figure 2a also shows that the global mean CH$_3$CCl$_3$ from the NOAA and AGAGE networks
differ by ~2.5ppt around 1993-1996, but since then this difference has become smaller.
The observed and modelled CH$_3$CCl$_3$ decay rate anomalies (calculated using the method of
Holmes et al., (2013) with a 12-month smoothing) are shown in Figures 2b and 2c (different
panels are used for AGAGE and NOAA comparisons for clarity). The model and observation-



derived results both tend to show a faster $CH_3CCl_3$ decay (more positive anomaly) in the middle
of the period, with slower decay at the start and end. The anomalies for the NOAA and
AGAGE-derived OH show periodic variations on a timescale of 2-3 yrs but with a phase shift
between the two datasets of 3 months, as noted by Holmes et al., (2013). The model runs with
OH variability prescribed from the observations and varying winds also show these periodic
variations with correlation coefficients ranging from 0.71 – 0.90. The correlation values for
these runs using varying OH are all larger than the run using repeating OH (for RE_FTVW
R=0.62 compared to AGAGE data and 0.67 compared to NOAA data). Note that for $CH_3CCl_3$
decay there are only small differences between the 3-D simulations which use varying
temperatures and the corresponding runs which use fixed temperature (e.g. simulation
RE_VTVW versus RE_FTVW). This agrees with the results of Montzka et al (2011) based on
their box model. This shows that the largest contribution from the $CH_3CCl_3$ decay rate anomaly
comes from variations in atmospheric OH concentration, rather than atmospheric temperature.
The simulations with repeating winds show less variability in the $CH_3CCl_3$ decay rate,
particularly in the period 1999-2004, but the small difference suggests that the interannual
variability in the observed $CH_3CCl_3$ decay rate is driven primarily by the variations in the OH
concentration. The remaining interannual variability in run RE_FTFW is due to variations in
emissions.
Figure 3 shows the $CH_3CCl_3$ decay and decay rate anomalies at four selected stations, two from
the NOAA network and two from the AGAGE network. The good agreement in the global
$CH_3CCl_3$ decay in Figure 2 is also seen at these individual stations. At the AGAGE stations of
Mace Head and Gape Grim, the model runs with varying OH perform better in capturing the
decay rate anomalies than the runs with repeating OH. However, the impact of variability in
the winds (solid lines versus dotted lines) is more apparent at these individual stations
compared to the global means. At the NOAA station of Mauna Loa the model run with varying
OH and varying winds also appears to perform better in capturing the observed variability in
$CH_3CCl_3$ decay. At the South Pole the observed variability is small, except in 2000-2002. This
feature is not captured by the model.
In summary, Figures 2 and 3 show that the global OH fields that we have constructed from
different datasets can perform well in capturing the decay of $CH_3CCl_3$ and its anomalies both
globally and at individual stations. Although, the interannual variability in global mean OH has
been derived from these $CH_3CCl_3$ observations, the figures do show that the reconstructed
model OH fields (which also depend on the methodology discussed in Section 2) perform well
in simulating $CH_3CCl_3$ within the 3D model. Therefore, we would argue that these fields are
suitable for testing the impact of OH variability on the methane growth rate. Even so, it is
important to bear in mind that these fields may not represent the true changes in atmospheric
OH, particularly if the interannual variability in $CH_3CCl_3$ emissions was a lot different to that
assumed here. However, we would again note that we are focussing on the impact of multi-
year variability which appears more robustly determined by the networks under differing
assumptions of temperature and emissions than year-year variability.

**3.2.2 Methane**



Figure 4 shows deseasonalised modelled surface $CH_4$ from the 3-D CTM simulations compared
with in-situ observations from a northern high-latitude station (Alert), two tropical stations
(Mauna Loa and Tutuila), a southern high-latitude station (South Pole) and the global average
of the NOAA and AGAGE stations. The global comparisons are shown for simulations both
with varying and repeating meteorology. Figure 5 shows the global annual $CH_4$ growth rates
with a 12-month smoothing (panel a) and differences between the model and NOAA and
AGAGE observations (panels b and c). The changes in the modelled global mean $CH_4$ over
different time periods are given in Table 3.
Figure 4 shows that in 1993, at the end of the model spin-up, the simulations capture the global
mean $CH_4$ level well, along with the observed values at a range of latitudes. The exception is
at high northern latitudes. However, these differences are not important when investigating the
change in the global growth rate. The global change in atmospheric $CH_4$ in the simulations
with varying winds for 1993 to the end of 2009 is between 75 and 104 ppb, compared to 56
and 66 ppb in the observations.
Model run RE_FTFW does not include interannual variations in atmospheric transport or $CH_4$
loss. Therefore, the modelled $CH_4$ gradually approaches a steady state value of ~1830 ppb
(Figure 4f). The rate of $CH_4$ growth decreases from 7.9 ppb/yr (1993-1998) to 1.4 ppb/yr (2007-
2009). Compared to run RE_FTFW, the other simulations introduce variability on this $CH_4$
evolution.
Run RE_FTVW includes interannual variability in wind fields which may alter the transport
of $CH_4$ from the source (emission) to the sink regions. The largest difference between runs
RE_FTFW and RE_FTVW occurs after 2000 (Figure 4f). During the stagnation period (1999-
2006) run RE_FTVW has a smaller growth rate of 3.5 ppb/yr compared to 4.1 ppb/yr in run
RE_FTFW, showing that variations in atmospheric transport made a small contribution to the
slowdown in global mean $CH_4$ growth.
Compared to run RE_FTVW, runs AP_FTVW, AL_FTVW, NP_FTVW and NL_FTVW
include $CH_3CCl_3$-derived interannual variations in [OH] which introduce large changes in
modelled $CH_4$, which are more in line with the observations (Figure 4e and 5). These runs
produce turnarounds in the $CH_4$ growth in 2001/2 (becomes negative) and 2005/6 (returns to
being positive). For AGAGE-derived [OH] (runs AP_FTVW, AL_FTVW) the large negative
anomaly in OH in 1997 produces a significant increase in $CH_4$ prior to the turnround in 2001.
Table 3 summarises the change in global mean $CH_4$ over different time periods. These periods
are defined by the key dates in the observed record, i.e. 1999 and 2006 as the start and end
dates of the stagnation period. Comparison of Figure 4e and Table 3 shows, however, that the
timing of the largest modelled change in growth rate do not necessarily coincide with those
dates. That is understandable if other factors not considered here, e.g. emission changes, are
contributing to the change in global $CH_4$ concentration. It does mean that the summary values
in Table 3 do not capture the full impact of the changes in [OH] and winds within the stagnation
period. Figure 4e shows that model runs with varying OH perform better in simulating the
relative CH4 trend 1999 to around 2004.



Table 3 shows that runs NP_FTVW and NL_FTVW (NOAA-derived [OH]) produce a small
modelled $CH_4$ growth of 2.5-3.1 ppb/yr during the stagnation period 1999-2006, compared to
1.0 ppb/yr for run AP_FTVW (AGAGE-derived [OH]). The AGAGE results are slightly larger
than the observed growth rate of 0.6-0.7 ppb/yr. Runs AL_FTVW, AP_FTVW, NL_FTVW
and NP_FTVW capture the observed strong decrease in the $CH_4$ growth rate. Clearly, these
runs demonstrate the significant potential for relatively small variations in mean [OH] to affect
$CH_4$ growth. Excluding the stagnation period the mean modelled $CH_4$ lifetime in run
NP_FTVW is 9.4 years, but this decreases slightly by 0.01 years during the stagnation period.
For run AP_FTVW there is a decrease of 0.18 years from 9.6 years between the same intervals.
The results from all the CTM simulations during 1999-2006 indicate that the accuracy of
modelled $CH_4$ growth is improved by accounting for interannual variability in [OH] as derived
from $CH_3CCl_3$ observations, and interannual variability in meteorology.
The variation of [OH] after 2007 cannot be determined from the available NOAA data so run
NP_FTVW used the mean [OH] field for all subsequent years. The modelled $CH_4$ increase of
3.5 ppb/yr underestimates the observations (4.9 ppb/yr). Should the lower [OH] of 2007 have
persisted then the model would have produced a larger increase in $CH_4$, in better agreement
with the observations. The AGAGE-derived [OH] for 2007-2009 (run AP_FTVW) produces a
larger $CH_4$ growth relative to the previous years (8.8 ppb/yr). Runs RE_FTFW (1.4 ppb/yr) and
RE_FTVW (1.8 ppb/yr) both show a decreased rate of growth during the final 5 years,
consistent with a system approaching steady state.
Figure 5a shows the global $CH_4$ growth rate derived from the AGAGE and NOAA networks
together with selected model simulations. Figures 5b and c show the differences between the
model simulations and the NOAA and AGAGE observations, respectively. The runs which
include variations in [OH] agree better with the observed changes, i.e. larger R values in panel
(a) and the model lines are closer to the y=0 line in panels (b) and (c), especially in the first 5
years of the stagnation period. It is interesting to note that the relative impacts of wind and
temperature variations are larger for $CH_4$ than for $CH_3CCl_3$ (compare simulations RE_FTFW,
RE_FTVW and RE_VTVW in Figures 2 and 5a). The temperature dependences of the OH loss
reactions are similar for the two species (see Supplement S1) but the impact of transport from
emissons regions to chemical loss regions is more variable for $CH_4$. This needs to be considered
when applying results derived from $CH_3CCl_3$ to $CH_4$.

## 350  4. Discussion and Conclusions

Our model results suggest that variability in atmospheric [OH] and transport played key roles
in the observed recent variations in $CH_4$ growth, particularly during the $CH_4$ stagnation period
between 1999 and 2006. The 3-D CTM calculations show that during the stagnation period,
variations in atmospheric conditions in the tropical lower to mid-troposphere could potentially
account for an important component of the observed decrease in global $CH_4$ growth. Within
this, small increases in [OH] were the largest factor, while variations in transport made a
smaller contribution. Note again, however, that the ultimate loss of $CH_4$ is still due to
chemistry. The role of atmospheric temperature variations is factored into the observationally
derived OH, but model experiments show that changes in the OH concentration itself is most




important. The remainder of the variation can be ascribed to other processes not considered in
our runs such as emission changes. There are also measurement uncertainties to consider and
the possible underrepresentation of the global mean $CH_3CCl_3$ which will affect the derived OH
concentration. Our results are consistent with an earlier budget study which analysed 1991 to
2004 and found that variations in [OH] were the main control of variations in atmospheric $CH_4$
lifetime (65%), with temperature accounting for a smaller fraction (35%) (Fiore et al., 2006).
As we have noted here the $CH_4$ lifetime can also be affected by emissions distributions which
affects transport to the main loss regions.
Prior to the stagnation period the simulation using AGAGE-derived [OH] overestimates $CH_4$
growth when compared to observations which degrades the agreement with the observed $CH_4$
variations. A likely cause of this is inaccuracies in derived [OH] in 1997 when emissions still
played a large role in the observed $CH_3CCl_3$ and the e-fold decay had not yet stabilised
(Montzka et al., 2011).
We have not accounted for expected variations in $CH_4$ emissions in this study. We can conclude
that although global $CH_4$ emissions do vary year-to-year, the observed trend in $CH_4$ growth
between 1999 and 2006 was impacted by changing atmospheric processes that affected $CH_4$
loss. Changes in emissions are still important and likely still dominate $CH_4$ variations over
other time periods. The observed changes in growth rates during ENSO events in e.g. 1998 are
poorly captured by the meteorological changes considered here and can be attributed to changes
in emissions through changing precipitation and enhanced biomass burning (Hodson et al.,
2011). The renewed growth of $CH_4$ in 2007 is also poorly captured by all model simulations
without varying [OH]. The observed decrease in AGAGE and NOAA-derived [OH] coincides
with the increase in $CH_4$ growth in 2007, although the currently available data do not allow for
a more detailed investigation of the possible contribution of [OH] changes in this recent
increase.
Despite the differences in year-to-year variability in [OH] derived from $CH_3CCl_3$ observations
(Holmes et al., 2013), we find that [OH] variability derived from two different networks of
surface $CH_3CCl_3$ observations over multi-year periods provide insights into atmospheric $CH_4$
variations. Improved quantification of the role of OH variability will require efforts to reduce
uncertainties associated with estimating [OH]. Estimates of global mean [OH] in recent years
from $CH_3CCl_3$ observations is becoming increasingly difficult because $CH_3CCl_3$ levels are
currently <5 ppt; hence this may limit the accuracy of derived [OH] and its variability in future
years (Lelieveld et al., 2006). Wennberg et al. (2004) also noted that there can be time
variations in the small uptake of $CH_3CCl_3$ by the oceans, which can also affect the derived
[OH] concentrations and are not considered here. Overall our study suggests that future
atmospheric trends in $CH_4$ are likely to be strongly influenced by not only emissions but also
changes in processes that affect atmospheric loss. The accuracy of predictions would therefore
be improved by including variations in [OH] and meteorology.
**Acknowledgements:** JRM thanks NERC National Centre for Earth Observation (NCEO) for
a studentship. CW, MPC and MG acknowledge support from NERC grants GAUGE
(NE/K002244/1) and AMAZONICA (NE/F005806/1). GDH acknowledges support from the





European Space Agency through its Support to Science Element initiative (ALANIS Methane),
NCEO and the NERC MAMM grant (NE/I028327/1). SAM acknowledges support in part from
NOAA Climate Program Office's AC4 program. AGAGE is supported by NASA grants
NNX11AF17G to MIT and NNX11AF15G and NNX11AF16G to SIO, by NOAA, UK
Department of Food and Rural Affairs (DEFRA) and UK Department for Energy and Climate
Change (DECC) grants to Bristol University, and by CSIRO and Australian Bureau of
Meteorology. MR is supported by a NERC Advanced Fellowship (NE/I021365/1). Model
calculations were performed on the Arc1 and Archer supercomputers.




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





**Tables**

**Table 1**. List of NOAA and AGAGE stations which provided $CH_4$ and $CH_3CCl_3$ observations.

| Site Code | Site Name | Lat. (ºN) | Lon. (ºN) | Altitude (km) | $CH_4$ | $CH_3CCl_3$ | Start Date[++] | End Date |
|---|---|---|---|---|---|---|---|---|
| ABP | Arembepe, Brazil | -12.77 | -38.17 | 0 | NOAA | | 27/10/2006 | 12/01/2010 |
| ALT | Alert, Canada | 82.45 | -62.51 | 0.2 | NOAA | NOAA | 10/06/1985 | Ongoing |
| ASC | Ascension Island, UK | 7.97 | -14.4 | 0.09 | NOAA | | 11/05/1983 | Ongoing |
| BRW | Barrow, USA | 71.32 | -156.61 | 0.01 | NOAA | NOAA | 06/04/1983 | Ongoing |
| CGO | Cape Grim, Australia | -40.68 | 144.69 | 0.09 | NOAA/AGAGE | NOAA/AGAGE | 19/04/1984 | Ongoing |
| HBA | Halley Station, UK | -75.61 | -26.21 | 0.03 | NOAA | | 17/01/1983 | Ongoing |
| ICE | Storhofdi, Iceland | 63.4 | -20.29 | 0.12 | NOAA | | 02/10/1992 | Ongoing |
| KUM | Cape Kumukahi, USA | 19.5 | -154.8 | 0.02 | - | NOAA | - | - |
| LEF | Park Falls, USA | 45.9 | -90.3 | 0.47 | - | NOAA | - | - |
| MHD | Mace Head, Ireland | 53.33 | -9.9 | 0.01 | NOAA/AGAGE | AGAGE** | 03/06/1991 | Ongoing |
| MLO | Mauna Loa, USA | 19.54 | -155.58 | 3.4 | NOAA | NOAA | 06/05/1983 | Ongoing |
| NWR | Niwot Ridge, USA | 40.05 | -105.59 | 3.52 | NOAA | NOAA | 21/06/1983 | Ongoing |
| PAL | Pallas-Sammaltunturi, Finland | 67.97 | 24.12 | 0.56 | NOAA | | 21/12/2001 | Ongoing |
| PSA | Palmer Station, USA | -64.92 | -64 | 0.01 | NOAA | ** | 01/01/1983 | Ongoing |
| RPB | Ragged Point, Barbados | 13.17 | -59.43 | 0.02 | NOAA/AGAGE | AGAGE | 14/11/1987 | Ongoing |
| SEY | Mahe Island, Seychelles | -4.68 | 55.53 | 0 | NOAA | | 12/05/1983 | Ongoing |
| SMO | Tutuila, American Samoa | -14.25 | -170.56 | 0.04 | NOAA | NOAA/AGAGE | 23/04/1983 | Ongoing |
| SPO | South Pole, USA | -89.98 | -24.8 | 2.81 | NOAA | NOAA | 20/02/1983 | Ongoing |
| STM | Ocean Station M, Norway | 66 | 2 | 0 | NOAA | | 29/04/1983 | 27/11/2009 |
| SUM | Summit, Greenland | 72.6 | -38.42 | 3.21 | NOAA | ** | 23/06/1997 | Ongoing |
| THD | Trinidad Head, USA | 41.1 | -124.1 | 0.1 | AGAGE | AGAGE** | 09/1995 | Ongoing |
| ZEP | Ny-Alesund, Norway/Sweden | 78.91 | 11.89 | 0.47 | NOAA | | 11/02/1994 | Ongoing |


++For NOAA $CH_3CCl_3$ data the record starts in 1992 at 7 of the 9 stations used here. It started
in 1995 for KUM and 1996 for LEF.
**NOAA flask data from these sites was not used in the present study or in Montzka et al.,

(2011).



**Table 2**. Summary of the fifteen TOMCAT 3-D CTM simulations.

| Run | OH time variation | Meteorology[b] | |
|-----|-------------------|-------|-------|
|     |                   | Winds[c] | Temperature[d] |
| RE_FTFW | Repeating[a] | Fixed | Fixed |
| RE_FTVW | Repeating[a] | Varying | Fixed |
| RE_VTVW | Repeating[a] | Varying | Varying |
| AP_FTFW | AGAGE (Rigby et al., 2013) | Fixed | Fixed |
| AP_FTVW | AGAGE (Rigby et al., 2013) | Varying | Fixed |
| AP_VTVW | AGAGE (Rigby et al., 2013) | Varying | Varying |
| AL_FTVT | AGAGE (this work) | Fixed | Fixed |
| AL_FTVW | AGAGE (this work) | Varying | Fixed |
| AL_VTVW | AGAGE (this work) | Varying | Varying |
| NP_FTFW | NOAA (Montzka et al., 2011) | Fixed | Fixed |
| NP_FTVW | NOAA (Monztka et al., 2011) | Varying | Fixed |
| NP_VTVW | NOAA (Montzka et al., 2011) | Varying | Varying |
| NL_FTFW | NOAA (this work) | Fixed | Fixed |
| NL_FTVW | NOAA (this work) | Varying | Fixed |
| NL_VTVW | NOAA (this work) | Varying | Varying |

a. Annually repeating [OH] taken from Patra et al. (2011).
b. Varying winds and temperatures are from ERA-Interim.
c. Fixed winds using repeating ERA-Interim winds from 1996.
d. Fixed temperatures use zonal mean ERA-Interim temperatures averaged over 1993-2009.



**Table 3**. Calculated methane changes over different time periods from selected TOMCAT
experiments and the NOAA and AGAGE observation networks.

| Model run or observation network | Global mean $\Delta CH_4$ /ppb (ppb/yr) | | | |
|---|---|---|---|---|
| | 2009-1993 | 1998-1993 | 2006-1999 | 2009-2007 |
| RE_FTFW | 85.0 (5.0) | 47.2 (7.9) | 32.9 (4.1) | 4.3 (1.4) |
| RE_FTVW | 82.2 (4.8) | 48.2 (8.0) | 27.8 (3.5) | 5.4 (1.8) |
| RE_VTVW | 74.6 (4.4) | 45.6 (7.6) | 23.1 (2.9) | 5.3 (1.8) |
| AP_FTVW[a] | 97.7[e] (5.7) | 62.3[e] (10.4) | 8.2 (1.0) | 26.4 (8.8) |
| AL_FTVW[b] | 104.2[e] (6.1) | 58.4[e] (9.7) | 17.3 (2.2) | 27.5 (9.2) |
| NP_FTVW[c] | 86.2[f] (5.1) | 49.7[f] (8.3) | 24.8 (3.1) | 10.6[f] (3.8) |
| NL_FTVW[d] | 91.4[f] (5.4) | 58.8[f] (9.8) | 20.1 (2.5) | 11.3[f] (3.8) |
| | | | | |
| NOAA obs. | 56.1 (3.3) | 36.0 (6.0) | 4.8 (0.6) | 14.7 (4.9) |
| AGAGE obs. | 66.3 (3.9) | 42.6 (7.1) | 5.6 (0.7) | 17.4 (5.8) |

a. Taken from Rigby et al. (2013) and Patra et al. (2011).
b. Using 1997-2009 relative annual changes in mean [OH] derived from AGAGE data
(Cunnold et al., 2002).
c. Taken from Montzka et al. (2011) and Patra et al. (2011).
d. Using 1997-2007 relative annual changes in mean [OH] derived from NOAA data (Prinn
et al., 2015).
e. Value using mean [OH] from 1993-1996.
f. Value using mean [OH] from 1993-1996 and 2008-2011.





**Figures**

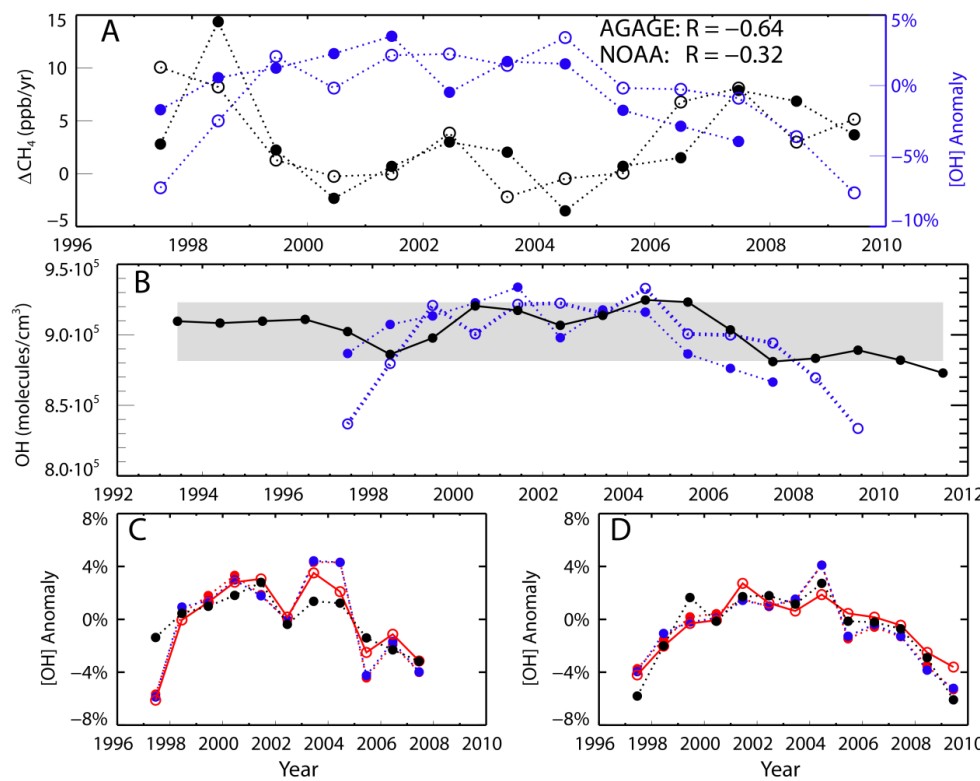


**Figure 1.** (a) Annual global $CH_4$ growth rate (ppb/yr) derived from NOAA (filled black circles)
and AGAGE (open black circles) data (left hand y-axis), and published annual global [OH]
anomalies derived from NOAA (filled blue circles, 1997-2007) and AGAGE (open blue
circles, 1997-2009) $CH_3CCl_3$ measurements (right hand y-axis) (see text). (b) Annual mean
[OH] (molecules/$cm^3$) required for global box model (see Supplement S1) to fit yearly
variations in NOAA $CH_4$ observations assuming constant emissions and temperature ($E$=553
Tg/yr, $T$=272.9 K), based on Montzka et al. (2011) (solid black line). The shaded region
denotes [OH] deviation of ±2.3% from the 1993-2011 mean. Also shown are the NOAA- and
AGAGE-derived anomalies from panel (a) for an assumed mean OH (see Section 2.1). (c) Our
estimates of [OH] derived from NOAA $CH_3CCl_3$ calculated using a global box model
(Supplement S1) using repeating (blue) and varying (red) annual mean temperature and the
$CH_3CCl_3$ emission scenario from UNEP (2015) (filled circles and dashed lines). Also shown
for varying temperatures are results using the emissions of Montzka et al (2011) (red open
circles and solid line) based on (Prinn et al. 2005) and the NOAA-derived values from panel
(a) (black dashed line and circles). (d) As panel (c) but for OH derived from AGAGE $CH_3CCl_3$
observations.




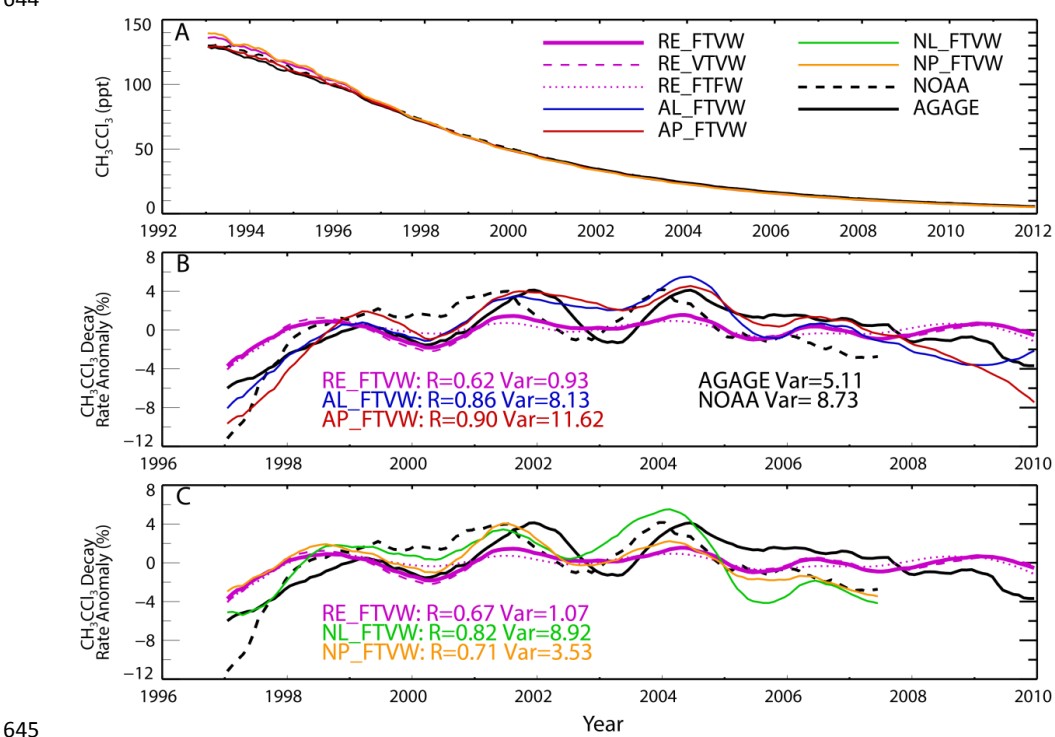


**Figure 2.** (a) Global mean surface $CH_3CCl_3$ (ppt) from NOAA (black dashed) and AGAGE (black solid) observations from 1993 to 2012. Also shown are results from five TOMCAT simulations with fixed temperatures and varying winds (see Table 1). (b) Global surface $CH_3CCl_3$ decay rate anomalies from NOAA and AGAGE along with model runs RE_FTVW, AL_FTVW and AP_FTVW (solid lines). Results from runs RE_FTFW and RE_VTVW are shown as a purple dotted line and dashed line, respectively. Observation and model anomalies are smoothed with a 12-month running average. Values given represent correlation coefficient when compared to AGAGE observations and variance. The decay rate anomaly is calculated from global mean $CH_3CCl_3$ values using equation (1) from Holmes et al., (2013), expressed as a percentage of the typical decay with a 12-month smoothing. (c) As panel (b) but for model runs NL_FTVW and NP_FTVW, along with RE_FTVW, RE_VTVW and RE_FTFW, and correlation cofficients for comparison with NOAA observations. The model results are split across panels (b) and (c) for clarity.





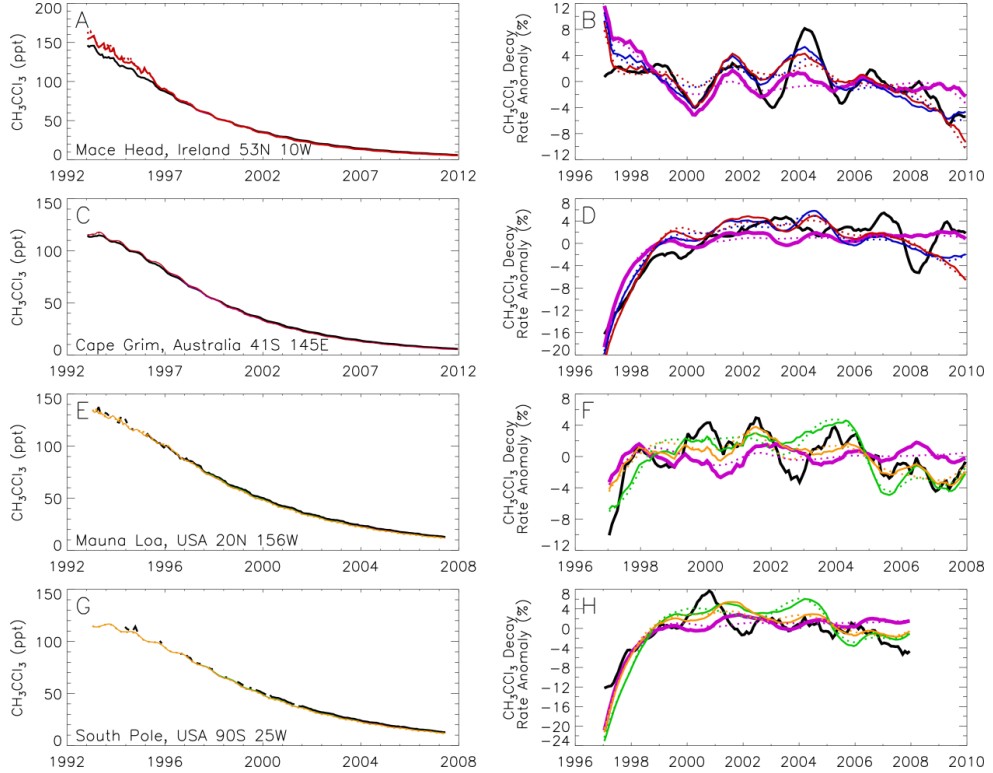

**Figure 3.** (Left) Observed mean surface CH$_3$CCl$_3$ (ppt) (black line) from (a) Mace Head (AGAGE), (c) Cape Grim (AGAGE), (e) Mauna Loa (NOAA) and (g) South Pole (NOAA). Also shown are results from five TOMCAT simulations with fixed temepartures and varying winds (FTVW, for legend see Figure 2a). (Right): Surface CH$_3$CCl$_3$ decay rate anomalies at the same station as the corresponding left column plot for observations (black), TOMCAT simulations with varying winds (FTVW, solid coloured lines) and TOMCAT simulations with fixed winds (FTFW, dotted lines). Comparisons at NOAA (AGAGE) stations show only comparisons with runs using NOAA (AGAGE)-derived OH, along with runs RE_FTVW and RE_FTFW in all panels.





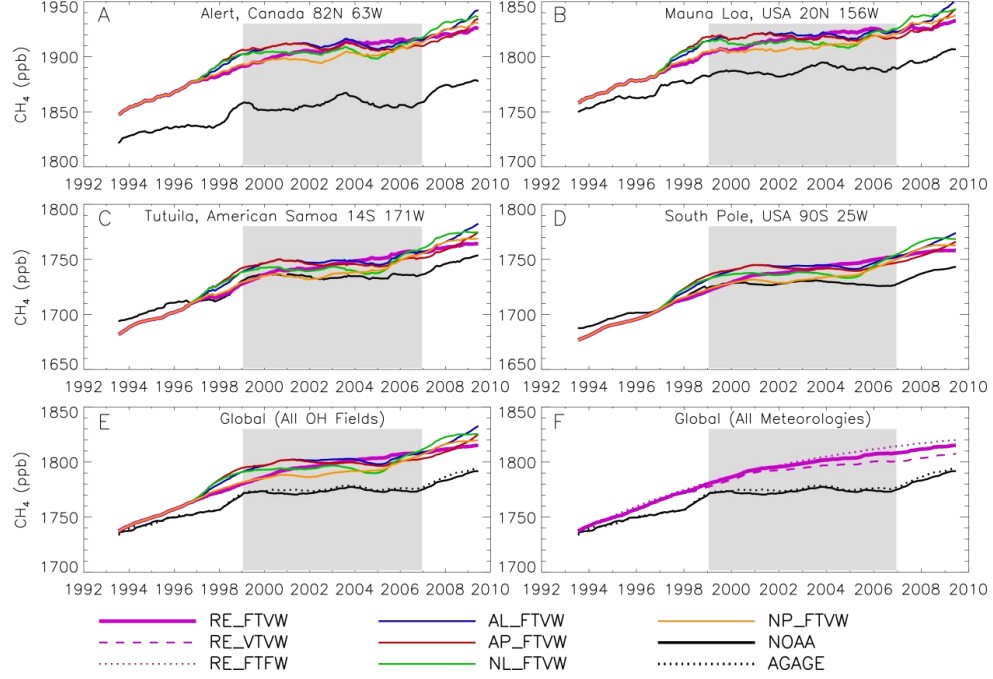

669

**Figure 4.** (a, b, c and d) Deasonalised surface CH$_4$ (ppb) from 4 NOAA sites (black solid line) from 1993 to 2009. Also shown are results from five TOMCAT 3-D CTM simulations with fixed temperatures and varying winds (FTVW, see **Table 2**). (e) Deasonalised global mean surface CH$_4$ from NOAA (black solid) and AGAGE (black dashed) observations along with five TOMCAT simulations with different treatments of OH. (f) Same as (e) but for TOMCAT simulations using repeating OH (RE) and different treatments of winds and temperature. All panels use observation and model values which are smoothed with a 12-month running average. The shaded region marks the stagnation period in the observed CH$_4$ growth rate.



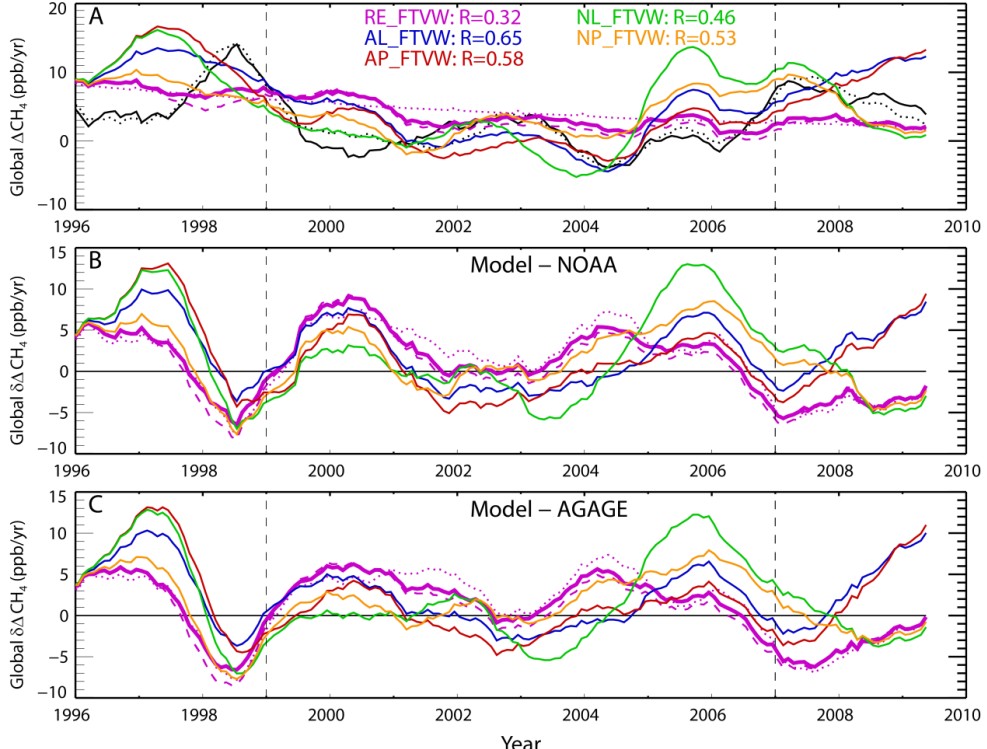

678

**Figure 5.** (a) The smoothed variation in the global annual $CH_4$ growth rate (ppb/yr) derived
from NOAA (black solid) and AGAGE (black dashed) observations. Also shown are the
smoothed growth rates from five TOMCAT 3-D CTM simulations with fixed temperatures and
varying winds (FTVW, see Table 1). Values in legend give correlation coefficient between
model run and NOAA observations. Also shown are results from runs RE_FTFW and
RE_VTVW as a purple dotted line and dashed line, respectively (b) The difference in smoothed
growth rate between TOMCAT simulations and NOAA observations shown in panel (a). (c)
Same as (b) except using differences compared to AGAGE observations. The vertical dashed
lines mark the start and end of the stagnation period in the observed $CH_4$ growth rate (1999 –
2006).