# Peer review of "Role of OH variability in the stalling of the global atmospheric CH4 growth rate from 1999 to 2006"

_Atmospheric Chemistry and Physics, 2015_

## Referee Comment (RC1) · Anonymous Referee #1 · 2 Mar 2016

GENERAL COMMENTS AND MAJOR SPECIFIC COMMENTS

This manuscript by J. McNorton et al. describes a set of chemical transport model simulations of atmospheric $CH_4$ during the 1990s through 2000s that use specified OH fields and year-to-year OH anomalies derived from $CH_3CCl_3$ measurements by previous studies and by the authors. The authors conclude that OH variations could explain a significant portion of the observed changes in $CH_4$ growth rate, including a drop to near zero during 1999-2006, with smaller contributions to the trends from variations in atmospheric transport and temperature.

Overall, I think this manuscript meets basic requirements for a publishable paper and has some good qualities, though it is somewhat thin on content. In its current form, it is perhaps more suited as a "letter" rather than a full-length article. Some of the work reported in the paper is mostly a confirmation that the authors can reproduce the results reported previously by others, particularly the yearly global OH anomalies derived by the authors from $CH_3CCl_3$ using a box model. And in my judgement, the paper makes a relatively small contribution to the body of scientific work, given that much of the work is not original or especially innovative. For example, the investigators used an OH distribution and yearly anomalies calculated by others. Also, the effects of transport and temperature on global $CH_4$ loss have already been studied by others (e.g. the Warwick et al. (2002) and Fiore et al. (2006) papers cited in this paper), though perhaps not for the $CH_4$ "stagnation period" that the current paper focusses on. Despite the shortcomings, I think the paper could become more suited for publication in *ACP* if the authors address my comments, in the process increasing the content of the paper. I do think the authors have done a good job of performing sets of $CH_3CCl_3$ and $CH_4$ simulations that test various potential influences on $CH_4$ trends, displaying the results thoroughly in figures and tables, and being candid about caveats and limitations of the study.

One major specific comment is that I'm not convinced that the year-to-year variations in OH can be estimated with a high level of certainty from $CH_3CCl_3$ measurements, given various uncertainties in the modeling, including assumed emissions (especially when emissions were still significant prior to around 2000). The authors themselves acknowledge some discrepancies between their estimated OH anomalies and those of published studies (page 6, lines 208-216). Thus, I see the findings on the contribution of OH variability to $CH_4$ trends as somewhat speculative. The higher correlations of the varying-OH runs with the observed $CH_4$ growth compared to the repeating-OH run in Fig. 5 could be a coincidence. A related comment is that the sub-periods delineated in Table 3 for trend calculations are rather short, so that the trends may not be robust. I think providing significance levels (p-values) for the trends would be helpful.

The authors make some statements in different parts of the paper that are not supported by sufficient evidence. Below, I note places where additional information or sensitivity tests could strengthen the statements.

OTHER SPECIFIC COMMENTS

This study relies entirely on the interannual OH variations inferred from $CH_3CCl_3$ observations and does not consider the OH variations suggested by other methods, including bottom-up, photochemical model calculations and top-down estimates using alternative halocarbons. The authors justify their use of specified OH with a comment near the beginning of Section 3.2.1 that "models with interactive tropospheric chemistry can produce a large range in absolute global mean [OH]", but they do not discuss the interannual variations in OH produced by such models. Montzka et al. [2011] show the OH variations derived from a photochemical model calculation as well as from various halocarbons including $CH_3CCl_3$ and note some of the differences. I think the current paper could be strengthened by considering other methods and possibly doing some sensitivity tests to assess how robust the conclusions are in the face of differing estimates of OH variations.

Section 2.1: Estimated anomalies in global OH based on $CH_3CCl_3$ measurements may not be accurate when applied to $CH_4$ given the different spatial distributions of $CH_4$ and $CH_3CCl_3$ and, to a lesser extent, different temperature dependences of their reaction with OH. The authors state at the end of Section 3 (lines 348-349) that this needs to be considered, but they do not actually consider it in their analysis. They should at the least emphasize this caveat more in the paper and discuss its implications for their findings.

Line 172: The runs that allow temperature to vary interannually would seem to doubly apply the temperature effect, given that the OH anomalies already implicitly contain temperature variations. Could you justify this?

Lines 180-182: You could discuss to what extent could the causality actually be bidirectional, i.e. high $CH_4$ growth can sometimes result in low OH, so that OH isn't always the sole driver of the OH-$CH_4$ correlations.

Lines 368-372: I suggest making this statement more quantitative, i.e. how large are the underestimate of OH and the overestimate of $CH_4$ growth?

Lines 376-377: Your analysis hasn't ruled out the possibility of changes in emissions being important during the 1999-2006 time period as well. Furthermore, the picture is more complex than all $CH_4$ sources varying in the same direction; decreases in certain sources could compensate for increases in other sources.

Lines 389-392: Is this issue relevant to your analysis? If so, could you suggest what impact it might have on your results? And if it isn't relevant, you could omit the sentence.

Lines 392-394: Could you estimate how large of an effect this uncertainty might have on your results?

Lines 394-397: This statement is certainly true and important, although it is not new and insightful. I suggest improving the statement so that the paper ends on a stronger note.

Figure 5b-c: It's not clear to me from these plots that the runs with varying OH are in better agreement with observations than the run with repeating OH is. Perhaps you could also report the mean values of model minus observations over the different sub-periods.

MINOR COMMENTS

Lines 49-50: The post-2006 growth rate of ~6 ppb/yr cited here seems inconsistent with the 4.9 ppb/yr given in the abstract. Please reconcile.

Line 68: You should provide references for the statement that "the reasons for the renewed growth are also not fully understood."

Lines 74-75: You could include additional references such as Wang et al. (2004) and Karlsdottir and Isaksen (2000). The full references are:

Wang, J. S., J. A. Logan, M. B. McElroy, B. N. Duncan, I. A. Megretskaia, and R. M. Yantosca (2004), A 3-D model analysis of the slowdown and interannual variability in the methane growth rate from 1988 to 1997, Global Biogeochem. Cycles, 18, GB3011, doi:10.1029/2003GB002180.

Karlsdottir, S., and I. S. A. Isaksen (2000), Changing methane lifetime: Possible cause for reduced growth, Geophys. Res. Lett., 27, 93– 96.

Lines 75-77: In addition, you could explain that Wang et al. attributed the OH trend to a decrease in column $O_3$ amounts, and the modeled trend of Karlsdottir and Isaksen was due to changes in tropospheric pollutants.

Lines 79-81: I have not read those papers (Carn et al., 2015; Mills et al., 2015), but my understanding is that in the troposphere, volcanoes are a much less important aerosol source than human activities, and volcanic aerosols that reach the stratosphere actually promote ozone depletion and thus increased downward UV and OH. So I think the counter-intuitive conclusion of those papers needs some explanation here.

Lines 81-83: The citing of this Patra et al. paper is not really relevant to the discussion in the paragraph on OH trends, so it does not belong here.

Line 157: This sentence is somewhat confusing, as I initially thought it meant that 1977 emissions were used for the entire 15-year spin-up. I suggest that you specify that emissions from 1977 to 1992 were used for the spin-up.

Lines 315-317: I find this sentence unclear. Are you referring to the observed growth rates? Please clarify.

Line 351: You include "transport" in this sentence as playing a key role, but your results suggested a relatively minor role. Perhaps you could reword this sentence.

Table 3: I understand your usage of "/ppb" in the heading of the table but it's not clear; maybe replace "/" with "in" or ",".

---

## Referee Comment (RC2) · Anonymous Referee #2 · 3 Mar 2016

The manuscript of McNorton et al. investigates the role of OH in driving the recent evolution of methane, especially the observed decline of its growth rates in the first half of the 2000s. The conclusion is that OH may have been a key driver of this modification of the methane growth rate.

The manuscript is well written and well within the scope of ACP. Even though there have been some key studies investigating the topic of the methane growth stagnation, this is the first paper that thoroughly investigates the role of OH. This is achieved through a series of model experiments with carefully chosen set-ups. I do not have any major concerns, but there are some (mostly minor) suggestions that I list below which I believe will improve the manuscript. Following those, I expect that it will be ready for publication.

[Figure]

[Figure]

GENERAL COMMENTS:

- I would have expected some discussion towards the end of the paper ("Discussion and Conclusions" section) on why the previous studies that investigated this stagnation in methane growth did not come up with a similar conclusion when it comes to the role of OH. This is the new bit that this paper brings, and it needs to be understood why those conclusions were not reached before. Some brief additions to the final section commenting on this aspect would be useful.

- Since the simulations start at 1993, why would the spin-up be done for 1977 conditions? That must be creating some methane imbalance in 1993, and with methane's relatively long lifetime, this will still be there in 1997, when the period of major interest begins. I may be missing something, but even in that case, it probably means that this aspect shall be clarified better.

SPECIFIC COMMENTS:

Page 1, Lines 18-19: Sentence not very clear. How can something vary "on a timescale of many years", within two decades?

Page 1, Line 29: Please add "of" between "and" and "atmospheric".

Page 2, Line 49: 6ppb/yr: Number inconsistent with the abstract. Please correct the one that is wrong.

Page 2, Line 57: Please add "potential" between "second" and "explanation".

Page 2, Line 60: "much more uncertainty" is unclear – please say a bit more.

Page 2, Line 66-67: So, the decrease in wetland emissions mentioned earlier was abandoned as a hypothesis. This paragraph needs to be connected in a clearer way with the previous one.

Page 2, Line 71: Suggest adding "global mean" before "concentration", as this symbol ("[OH]") is used throughout the manuscript when referring to the global abundance.

Page 3, Lines 77-78: A recent paper by Voulgarakis et al. also included findings along these lines when it comes to the role of fires on OH variability, especially during El Nino events (see their Fig. 4c):

Voulgarakis, A., M.E. Marlier, G. Faluvegi, D.T. Shindell, K. Tsigaridis, and S. Mangeon, 2015: Interannual variability of tropospheric trace gases and aerosols: The role of biomass burning emissions. J. Geophys. Res. Atmos., 120, no. 14, 7157-7173, doi:10.1002/2014JD022926.

Page 3, Lines 103-110: Need to also remind the reader of the main finding of the Montzka et al. (2011) paper, i.e. the suggested small interannual variability of OH.

Page 3, Line 107: Suggest changing "this" to "that".

Page 4, Line 117: Suggest adding "global" between "yearly" and "anomalies".

Page 4, Line 127: Suggest changing "date" to "year".

Page 4, Line 152: What is meant by "scaled"? Please clarify.

Page 4, Lines 167-168: Why were zonal means of temperature used and not 3D data? That introduces one potential extra reason for differences between the runs, i.e. not just the fact that the temperature is fixed, but also that it is not 3D-varying. What is the impact of this?

Page 4, Lines 168-169: Suggest rephrasing to "We also derive our own OH anomalies from the anomaly in the...".

Page 5, Line 156: Need to clarify whether the specified OH field is comprised of zonal means or whether it varies with longitude. If the former, need to discuss the implications of the lack of longitudinal variations.

Page 6, Line 231: It should be 0.65 rather than 0.55.

Page 7, Lines 274-275: What is meant by "multi-year" here? Suggest specifying with a

parenthesis.

Page 7, Line 276: "year-year" -> "year-to-year".

Page 8, Line 290-291: Why are the simulations with varying winds singled-out?

Page 8, Line 294: Suggest adding "and also given the lack of change in emissions" after "Therefore,".

Table 3: It is not immediately clear what is meant in the parentheses next to the numbers. I suggest writing "Global mean $\Delta CH4$ in ppb" at the top row and "Global mean $\Delta CH4$ per year in ppb/yr" at the bottom row of the title of those columns.

Page 8, Lines 315-317: I am not sure what is meant by this sentence. May need to be expanded or reworded.

Page 8, Line 318: In "CH4" the "4" needs to be subscripted. Also, I think a "from" is missing before "1999".

Page 9, Lines 346-349: This is interesting. But why could that be. An explanation, even a speculative one, would be nice. Is it perhaps due to somewhat different emissions regions for the two constituents, leading to different efficiencies of transport to regions of maximum loss?

Page 9, Lines 357-358: I do not see why this sentence is needed.

Page 10, Line 369: Please add "," before "which".

––––––––––––––––––––––––––

---

## Author Comment (AC1) · 22 Apr 2016

**Role of OH variability in the stalling of the global atmospheric CH₄ growth rate from 1999 to 2006 by J. McNorton et al.**

**Response to Reviewers' Comments**

We thank the reviewers for their time and constructive comments. These comments are repeated below (in normal text) followed by our responses *(in blue italics)*.

**Anonymous Referee #1**

GENERAL COMMENTS AND MAJOR SPECIFIC COMMENTS

This manuscript by J. McNorton et al. describes a set of chemical transport model simulations of atmospheric CH4 during the 1990s through 2000s that use specified OH fields and year-to-year OH anomalies derived from CH3CCl3 measurements by previous studies and by the authors. The authors conclude that OH variations could explain a significant portion of the observed changes in CH4 growth rate, including a drop to near zero during 1999-2006, with smaller contributions to the trends from variations in atmospheric transport and temperature.

Overall, I think this manuscript meets basic requirements for a publishable paper and has some good qualities, though it is somewhat thin on content. In its current form, it is perhaps more suited as a "letter" rather than a full-length article. Some of the work reported in the paper is mostly a confirmation that the authors can reproduce the results reported previously by others, particularly the yearly global OH anomalies derived by the authors from CH3CCl3 using a box model. And in my judgement, the paper makes a relatively small contribution to the body of scientific work, given that much of the work is not original or especially innovative. For example, the investigators used an OH distribution and yearly anomalies calculated by others. Also, the effects of transport and temperature on global CH4 loss have already been studied by others (e.g. the Warwick et al. (2002) and Fiore et al. (2006) papers cited in this paper), though perhaps not for the CH4 "stagnation period" that the current paper focusses on. Despite the shortcomings, I think the paper could become more suited for publication in *ACP* if the authors address my comments, in the process increasing the content of the paper. I do think the authors have done a good job of performing sets of CH3CCl3 and CH4 simulations that test various potential influences on CH4 trends, displaying the results thoroughly in figures and tables, and being candid about caveats and limitations of the study.

*We thank the reviewer for his/her detailed review and we will make changes to the manuscript accordingly. We acknowledge that we have we have used OH anomalies calculated by others but we wanted to use the published data where available. By also using our own box model we were able to investigate differences between the two published OH anomaly datasets which were produced by different methods and based on different CH3CCl3 observations. Although the other studies noted above did look at transport and temperature effects, they did not look at the CH4 stagnation period which is of high current scientific interest.*

One major specific comment is that I'm not convinced that the year-to-year variations in OH can be estimated with a high level of certainty from CH3CCl3 measurements, given various uncertainties in the modeling, including assumed emissions (especially when emissions were still significant prior to around 2000). The authors themselves acknowledge some discrepancies between their estimated OH anomalies and those of published studies (page 6, lines 208-216).

Thus, I see the findings on the contribution of OH variability to CH4 trends as somewhat speculative. The higher correlations of the varying-OH runs with the observed CH4 growth compared to the repeating-OH run in Fig. 5 could be a coincidence. A related comment is that the sub-periods delineated in Table 3 for trend calculations are rather short, so that the trends may not be robust. I think providing significance levels (p-values) for the trends would be helpful.

*We agree that uncertainties exist in the OH anomalies derived from $CH_3CCl_3$ measurements (e.g. lines 208-216), although we would argue, as others before us have, that uncertainties on emissions play a smaller role in deriving OH anomalies after 1997, which is the main period of interest here. Furthermore, the fact that the multi-year signals derived from both global CH3CCl3 measurement programmes are reasonably consistent adds some confidence in the signals being robust. Nevertheless, even with these caveats (which we acknowledge) we think that it is still important to point out this possible role of OH variations on the observed $CH_4$ trend.*

*We agree that the sub-periods are fairly short but they are determined by the periods over which the global mean $CH_4$ shows variations. This length does reduce the robustness of the trends but they are the periods we need to analyse. In the revised paper we will include significance levels as suggested.*

The authors make some statements in different parts of the paper that are not supported by sufficient evidence. Below, I note places where additional information or sensitivity tests could strengthen the statements.

OTHER SPECIFIC COMMENTS

This study relies entirely on the interannual OH variations inferred from CH3CCl3 observations and does not consider the OH variations suggested by other methods, including bottom-up, photochemical model calculations and top-down estimates using alternative halocarbons. The authors justify their use of specified OH with a comment near the beginning of Section 3.2.1 that "models with interactive tropospheric chemistry can produce a large range in absolute global mean [OH]", but they do not discuss the interannual variations in OH produced by such models. Montzka et al. [2011] show the OH variations derived from a photochemical model calculation as well as from various halocarbons including CH3CCl3 and note some of the differences. I think the current paper could be strengthened by considering other methods and possibly doing some sensitivity tests to assess how robust the conclusions are in the face of differing estimates of OH variations.

*We agree different species could be used in principle; however previous studies, e.g. Montzka et al. (2011), which used other chemical species to derive OH, conclude that $CH_3CCl_3$ measurements provide the most robust and independent estimates. Other species used in their study to derive OH anomalies have much larger budget uncertainties and therefore do not provide equally reliable estimates of OH when compared with $CH_3CCl_3$. Both HCFC and HFC emissions are in a high state of flux because some chemicals are being phased in and out, making them much less suitable for deriving reliable changes in OH.*

*We also acknowledge that long-term simulations of photochemical models could be used to derive OH anomalies. However, there currently exists large uncertainty in model-derived OH, as noted by the reviewer in reference to Voulgarakis et al. (2013). As noted by Montzka et al.*

*(2011) photochemical models (e.g. see Leliveld et al., 2004) typically suggest a smaller-interannual variability than CH₃CCl₃-derived OH even since 1998, suggesting the models may not be accurately representing processes governing OH concentrations. Given that they calculate very different mean values it is likely that they are missing processes and will calculate different interannual variations based on the ones that they do. We believe that investigating accuracies in bottom-up photochemical models is beyond the scope of this work. We will add some brief discussion to the paper.*

Section 2.1: Estimated anomalies in global OH based on CH3CCl3 measurements may not be accurate when applied to CH4 given the different spatial distributions of CH4 and CH3CCl3 and, to a lesser extent, different temperature dependences of their reaction with OH. The authors state at the end of Section 3 (lines 348-349) that this needs to be considered, but they do not actually consider it in their analysis. They should at the least emphasize this caveat more in the paper and discuss its implications for their findings.

*OK. As noted by Reviewer 2 this difference between CH4 and CH3CCl3 is interesting. We will add more discussion and caveats on this point. We feel that it is a small effect.*

Line 172: The runs that allow temperature to vary interannually would seem to doubly apply the temperature effect, given that the OH anomalies already implicitly contain temperature variations. Could you justify this?

*We realise that there is this 'double counting' and so we use the simulations with fixed model temperature (FT) in our main analysis (see lines 168-172). By also running the model with varying temperature we can diagnose the likely contribution of temperature variations on OH + CH4 rate (see lines 358-360), even if the model run itself (VTVW) is not the most realistic. Through this we see that the temperature effect is small. We will clarify this in the revised paper.*

Lines 180-182: You could discuss to what extent could the causality actually be bidirectional, i.e. high CH4 growth can sometimes result in low OH, so that OH isn't always the sole driver of the OH-CH4 correlations.

*OK. We will add a statement regarding this possible bidirectional effect based on available literature. However, even with a large change in CH₄ growth rate, the total CH₄ mixing ratio in the atmosphere does not change by much. Table 6 in Spivakovsky et al (2000) shows ~5% change in model CH₄ equates to ~1% change in model OH. A 5% change in CH₄ ( ~100ppb), far exceeds the annual growth changes observed, therefore we believe this change to be small.*

Lines 368-372: I suggest making this statement more quantitative, i.e. how large are the underestimate of OH and the overestimate of CH4 growth?

*OK. The values already provided in Table 3 will be inserted into the text to quantify this statement.*

Lines 376-377: Your analysis hasn't ruled out the possibility of changes in emissions being important during the 1999-2006 time period as well. Furthermore, the picture is more complex than all CH4 sources varying in the same direction; decreases in certain sources could compensate for increases in other sources.

*We agree and we have tried to be careful to acknowledge that variations in emissions may still play an important role (e.g. abstract line 30, line 373-, line 395). We will further clarify this where possible in the revised version.*

Lines 389-392: Is this issue relevant to your analysis? If so, could you suggest what impact it might have on your results? And if it isn't relevant, you could omit the sentence.

*Yes, this is relevant and is one reason why we cannot analyse the most recent years. For the results shown in the paper measurements were only used up until 2007 (NOAA) and 2009 (AGAGE), when the methylchloroform concentration was higher. The statement is made to address future issues with the use of [OH]. We will change this sentence to offer more clarification.*

Lines 392-394: Could you estimate how large of an effect this uncertainty might have on your results?

*We will look at the Wennberg et al paper and add some discussion. They comment that "the loss of methylchloroform to the oceans play a small but important role". The first order effect of the ocean was as a net sink as CH3CCl3 concentrations were increasing, and potentially a very small net source as concentrations were decreasing. However, the question is how large is the interannual variability in this small term? It is difficult to imagine that any interannual variation could be large enough to affect our conclusions. We will do some estimates with the box model based on assumed extreme variations in the Wennberg sink/source (which is further reason for us to be able to run our own box model as well as use published OH values – see earlier comment).*

Lines 394-397: This statement is certainly true and important, although it is not new and insightful. I suggest improving the statement so that the paper ends on a stronger note.

*OK. This will statement will be modified.*

Figure 5b-c: It's not clear to me from these plots that the runs with varying OH are in better agreement with observations than the run with repeating OH is. Perhaps you could also report the mean values of model minus observations over the different sub-periods.

*We agree that the plot in isolation makes the difference difficult to see, which is why Table 3 provides the growth values requested. The difference over the sub-periods can be read from there.*

MINOR COMMENTS

Lines 49-50: The post-2006 growth rate of ~6 ppb/yr cited here seems inconsistent with the 4.9 ppb/yr given in the abstract. Please reconcile.

*OK. This will be corrected to 4.9 ppb/yr for both and the "post-2006" will be changed to 2006-2009.*

Line 68: You should provide references for the statement that "the reasons for the renewed growth are also not fully understood."

*OK, references will be included.*

Lines 74-75: You could include additional references such as Wang et al. (2004) and Karlsdottir and Isaksen (2000). The full references are:
Wang, J. S., J. A. Logan, M. B. McElroy, B. N. Duncan, I. A. Megretskaia, and R. M. Yantosca (2004), A 3-D model analysis of the slowdown and interannual variability in the methane growth rate from 1988 to 1997, Global Biogeochem. Cycles, 18, GB3011, doi:10.1029/2003GB002180.
Karlsdottir, S., and I. S. A. Isaksen (2000), Changing methane lifetime: Possible cause for reduced growth, Geophys. Res. Lett., 27, 93– 96.

*OK. These references will be added.*

Lines 75-77: In addition, you could explain that Wang et al. attributed the OH trend to a decrease in column O3 amounts, and the modeled trend of Karlsdottir and Isaksen was due to changes in tropospheric pollutants.

*OK, this information will be added.*

Lines 79-81: I have not read those papers (Carn et al., 2015; Mills et al., 2015), but my understanding is that in the troposphere, volcanoes are a much less important aerosol source than human activities, and volcanic aerosols that reach the stratosphere actually promote ozone depletion and thus increased downward UV and OH. So I think the counter-intuitive conclusion of those papers needs some explanation here.

OK. This papers were added for completeness but this effect has not been quantified. We will remove them.

Lines 81-83: The citing of this Patra et al. paper is not really relevant to the discussion in the paragraph on OH trends, so it does not belong here.

*OK, the text referring to this paper will be moved.*

Line 157: This sentence is somewhat confusing, as I initially thought it meant that 1977 emissions were used for the entire 15-year spin-up. I suggest that you specify that emissions from 1977 to 1992 were used for the spin-up.

*OK. This will be clarified.*

Lines 315-317: I find this sentence unclear. Are you referring to the observed growth rates? Please clarify.

*Theses lines refer to the modelled values. This will be clarified.*

Line 351: You include "transport" in this sentence as playing a key role, but your results suggested a relatively minor role. Perhaps you could reword this sentence.

*Yes. This will be corrected*

Table 3: I understand your usage of "/ppb" in the heading of the table but it's not clear; maybe replace "/" with "in" or ",".

*OK. This will be revised.*

---

## Author Comment (AC2) · 22 Apr 2016

**Role of OH variability in the stalling of the global atmospheric CH$_4$ growth rate from 1999 to 2006 by J. McNorton et al.**

**Response to Reviewers' Comments**

We thank the reviewers for their time and constructive comments. These comments are repeated below (in normal text) followed by our responses *(in blue italics)*.

**Anonymous Referee #2**

The manuscript of McNorton et al. investigates the role of OH in driving the recent evolution of methane, especially the observed decline of its growth rates in the first half of the 2000s. The conclusion is that OH may have been a key driver of this modification of the methane growth rate.

The manuscript is well written and well within the scope of ACP. Even though there have been some key studies investigating the topic of the methane growth stagnation, this is the first paper that thoroughly investigates the role of OH. This is achieved through a series of model experiments with carefully chosen set-ups. I do not have any major concerns, but there are some (mostly minor) suggestions that I list below which I believe will improve the manuscript. Following those, I expect that it will be ready for publication.

*We thank the reviewer for his/her careful reading of our manuscript and insightful comments.*

GENERAL COMMENTS:

I would have expected some discussion towards the end of the paper ("Discussion and Conclusions" section) on why the previous studies that investigated this stagnation in methane growth did not come up with a similar conclusion when it comes to the role of OH. This is the new bit that this paper brings, and it needs to be understood why those conclusions were not reached before. Some brief additions to the final section commenting on this aspect would be useful.

*We are not aware that other studies explicitly considered variations in OH when investigating the stagnation period. That is a motivation for this work. Lines 363-365 relate our study to a similar previous study (Fiore et al., 2006) but that study did not cover the stagnation period. We will add a few words to clarify these points.*

Since the simulations start at 1993, why would the spin-up be done for 1977 conditions? That must be creating some methane imbalance in 1993, and with methane's relatively long lifetime, this will still be there in 1997, when the period of major interest begins. I may be missing something, but even in that case, it probably means that this aspect shall be clarified better.

*The model is spun up from 1977 to get a reasonable spatial distribution by 1993, which does create a model-observation imbalance in 1993. This is corrected for by scaling the model 1993 global average CH4 concentration to observed data before reinitialising the model for the 1993-2009 simulation. We will clarify this in the revised version.*

SPECIFIC COMMENTS:

Page 1, Lines 18-19: Sentence not very clear. How can something vary "on a timescale of many years", within two decades?

*We agree the term "many years" could be confusing. In the revised version we will modify this to say that the variability is over multiple years, i.e. 2-5 years.*

Page 1, Line 29: Please add "of" between "and" and "atmospheric".

*OK.*

Page 2, Line 49: 6ppb/yr: Number inconsistent with the abstract. Please correct the one that is wrong.

*OK, the number should read 4.9 ppb/yr.*

Page 2, Line 57: Please add "potential" between "second" and "explanation".

*OK.*

Page 2, Line 60: "much more uncertainty" is unclear – please say a bit more.

*OK. This will be modified to say that the bottom-up and top-down estimates differed.*

Page 2, Line 66-67: So, the decrease in wetland emissions mentioned earlier was abandoned as a hypothesis. This paragraph needs to be connected in a clearer way with the previous one.

*We agree with the reviewer that this could be confusing. The paragraph will be changed to show consistency between paragraphs L51-61 and L62-68.*

Page 2, Line 71: Suggest adding "global mean" before "concentration", as this symbol ("[OH]") is used throughout the manuscript when referring to the global abundance.

*OK.*

Page 3, Lines 77-78: A recent paper by Voulgarakis et al. also included findings along these lines when it comes to the role of fires on OH variability, especially during El Nino events (see their Fig. 4c): Voulgarakis, A., M.E. Marlier, G. Faluvegi, D.T. Shindell, K. Tsigaridis, and S. Mangeon, 2015: Interannual variability of tropospheric trace gases and aerosols: The role of biomass burning emissions. J. Geophys. Res. Atmos., 120, no. 14, 7157-7173, doi:10.1002/2014JD022926.

*OK, this reference will be included.*

Page 3, Lines 103-110: Need to also remind the reader of the main finding of the Montzka et al. (2011) paper, i.e. the suggested small interannual variability of OH.

*OK, this will be included.*

Page 3, Line 107: Suggest changing "this" to "that".

*OK.*

Page 4, Line 117: Suggest adding "global" between "yearly" and "anomalies".

*OK.*

Page 4, Line 127: Suggest changing "date" to "year".

*OK.*

Page 4, Line 152: What is meant by "scaled"? Please clarify.

*More detail will be included; scaled emissions are taken from Ciais et al. (2014) top-down estimates.*

Page 4, Lines 167-168: Why were zonal means of temperature used and not 3D data? That introduces one potential extra reason for differences between the runs, i.e. not just the fact that the temperature is fixed, but also that it is not 3D-varying. What is the impact of this?

*We needed to create a time-averaged dataset (1993-2009) for the model run and so much of the 3-D variability would be averaged out anyway. We agree with the reviewer that using zonal mean fields might still influence the results; however, this difference is likely to be small (the major temperature variations, due to both height and latitude are considered). In any case we will add this note to the revised version.*

Page 4, Lines 168-169: Suggest rephrasing to "We also derive our own OH anomalies from the anomaly in the…".

*OK.*

Page 5, Line 156: Need to clarify whether the specified OH field is comprised of zonal means or whether it varies with longitude. If the former, need to discuss the implications of the lack of longitudinal variations.

*The field comprises of zonal means. A sentence will be included to discuss the implications of this.*

Page 6, Line 231: It should be 0.65 rather than 0.55.

*OK, corrected.*

Page 7, Lines 274-275: What is meant by "multi-year" here? Suggest specifying with a parenthesis.

*This will be modified to now include (>1 year).*

Page 7, Line 276: "year-year" -> "year-to-year".

*OK.*

Page 8, Line 290-291: Why are the simulations with varying winds singled-out?

*This is a miswording. The text will be changed to say that it considers all simulations not just those with varying winds.*

Page 8, Line 294: Suggest adding "and also given the lack of change in emissions" after "Therefore,".

*OK.*

Table 3: It is not immediately clear what is meant in the parentheses next to the numbers. I suggest writing "Global mean _CH4 in ppb" at the top row and "Global mean_CH4 per year in ppb/yr" at the bottom row of the title of those columns.

*OK, this will be clarified (see also comment from Reviewer 1).*

Page 8, Lines 315-317: I am not sure what is meant by this sentence. May need to be expanded or reworded.

*OK, this will be reworded (see also comment from Reviewer 1).*

Page 8, Line 318: In "CH4" the "4" needs to be subscripted. Also, I think a "from" is missing before "1999".

*OK.*

Page 9, Lines 346-349: This is interesting. But why could that be. An explanation, even a speculative one, would be nice. Is it perhaps due to somewhat different emissions regions for the two constituents, leading to different efficiencies of transport to regions of maximum loss?

*OK. We do not want to add unfounded speculation to the paper but we will try to expand on this result slightly in the revised version. A different in the spatial distribution of emissions would seem to be a potentially important factor.*

Page 9, Lines 357-358: I do not see why this sentence is needed.

*OK. We wanted to make the point that ultimately it is always chemistry that removed CH4 and not transport. We will clarify this.*

Page 10, Line 369: Please add "," before "which".

*OK*

---

## Author Response (AR2)

**Role of OH variability in the stalling of the global atmospheric CH4 growth rate from 1999 to 2006 by J. McNorton et al.**

**Response to Reviewer's Comments**

We thank the reviewer for his/her further time and comments. These comments are repeated below (in normal text) followed by our responses *(in blue italics)*.

**Reviewer 1**

Overall, I think the authors have addressed most of the comments in my first review. However, I have a few additional comments that I think are important to address:

In my comment about the robustness of the subperiod trends, I should have suggested calculating and reporting the standard errors (accounting for autocorrelation) rather than the statistical significance, or p-values, of the trends. The former indicates how well we know the trend, while the latter indicates whether a calculated trend is significantly different from zero, which is not exactly what we are interested in here. So apologies for the mistake, and please make the change.

*We agree the inclusion of standard errors is important to show the robustness of the calculated trends. We have updated Table 3 to include the standard errors for each of the subperiods and added text to the figure title. We used unsmoothed data. We tested the unsmoothed data for autocorrelation with a lag of multiple months and found no noticeable correlation and so no autocorrelation correction was applied to the final standard error calculation. We found that the size of the errors does not noticeably influence the conclusions made.*

I don't think you adequately addressed the comment from both myself and the other reviewer about anomalies in global OH based on CH3CCl3 measurements possibly not being accurate when applied to CH4. I see that you added a sentence at the end of Section 3, but it barely adds anything to the discussion.

The original comments were:

**Reviewer 1**: Section 2.1: Estimated anomalies in global OH based on CH3CCl3 measurements may not be accurate when applied to CH4 given the different spatial distributions of CH4 and CH3CCl3 and, to a lesser extent, different temperature dependences of their reaction with OH. The authors state at the end of Section 3 (lines 348-349) that this needs to be considered, but they do not actually consider it in their analysis. They should at the least emphasize this caveat more in the paper and discuss its implications for their findings.

**Reviewer 2**: Page 9, Lines 346-349: This is interesting. But why could that be. An explanation, even a speculative one, would be nice. Is it perhaps due to somewhat different emissions regions for the two constituents, leading to different efficiencies of transport to regions of maximum loss?

*We have expanded the text at the end of Section 3 and also added two figures and text into the Supplementary Material S2. In particular, by now showing the spatial gradients in $CH_4$ and $CH_3CCl_3$ (Figure S2) we illustrate how transport variability could have a larger effect on $CH_4$ than $CH_3CCl_3$. Moreover, this shows that it is better to derive OH from a 'well-mixed' species like $CH_3CCl_3$ in its period of decay than from $CH_4$ which has a lot of spatial variability. In that sense applying the OH variability derived from $CH_3CCl_3$ decay to $CH_4$ is better than the opposite. We think that this new information has*

*clarified the point we made in the original submission which led to the comments in the first review. Please see the new text and Supplement S2 for more information.*

Be sure to proofread your latest additions to the manuscript. For example, your statement in Section 3.1 on "a bidirectional effect" seems incomplete to me, since you discuss the potential impact of [CH4] on [OH] but never mention explicitly in that paragraph your assumption that OH is the primary driver of the correlation. Also, note that concentrations of CO and VOCs sometimes co-vary with [CH4], such as during years with high biomass burning, so the driving of the correlation in the direction of [CH4] to [OH] may be stronger than suggested by your rough estimate.

*OK, thank you. We have rewritten this part of the paper ('We assume that this correlation… study') to mention that we are assuming that OH drives CH$_4$, and that concentrations of OH and VOCs may co-vary with CH$_4$.*

*We also discovered that a preference was set inside our Word file to switch off the spell checker. We have now switched it on and discovered around 5 simple spelling errors.*